# Carbon Dots as Promising Tools for Cancer Diagnosis and Therapy

**DOI:** 10.3390/cancers13091991

**Published:** 2021-04-21

**Authors:** Giuseppe Nocito, Giovanna Calabrese, Stefano Forte, Salvatore Petralia, Caterina Puglisi, Michela Campolo, Emanuela Esposito, Sabrina Conoci

**Affiliations:** 1Department of Chemistry, Biology, Pharmacy and Environmental Science, University of Messina, 98122 Messina, Italy; giuseppe.nocito@unime.it (G.N.); campolom@unime.it (M.C.); eesposito@unime.it (E.E.); 2IOM Ricerca, Viagrande, 95029 Catania, Italy; stefano.forte@grupposamed.com (S.F.); caterina.puglisi@grupposamed.com (C.P.); 3Department of Drug Science and Health, University of Catania, 95125 Catania, Italy; salvatore.petralia@unict.it

**Keywords:** carbon dots, cancer diagnosis cancer therapy, bioimaging, drug delivery, photodynamic therapy, photothermal therapy

## Abstract

**Simple Summary:**

Diagnostic approaches and chemotherapeutic delivery based on nanotechnologies, such as nanoparticles (NPs), could be promising candidates for the new era of cancer research. Recently great attention has been received by carbon-based nanomaterials such as Carbon Dots (CDs), due their variegated physical-chemical properties that makes these systems appealing for multiple use from bioimaging, biosensing, nano-carriers for drug delivery systems to innovative therapeutic agents in photodynamic (PDT) and photothermal therapy (PTT). In this review, we report the last evidence on the application and prospects of CDs as useful nano theranostics tools for cancer diagnosis and therapy.

**Abstract:**

Carbon Dots (CDs) are the latest members of carbon-based nanomaterials, which since their discovery have attracted notable attention due to their chemical and mechanical properties, brilliant fluorescence, high photostability, and good biocompatibility. Together with the ease and affordable preparation costs, these intrinsic features make CDs the most promising nanomaterials for multiple applications in the biological field, such as bioimaging, biotherapy, and gene/drug delivery. This review will illustrate the most recent applications of CDs in the biomedical field, focusing on their biocompatibility, fluorescence, low cytotoxicity, cellular uptake, and theranostic properties to highlight above all their usefulness as a promising tool for cancer diagnosis and therapy.

## 1. Introduction

Cancer is one of the major health problems affecting the world population, with high incidence and mortality. It has been estimated that 9.6 million people died from cancer in 2017 and that every sixth death in the world is due to it, making it the second leading cause of death after cardiovascular disease [1]. Although significant advances have been achieved in cancer treatment during recent years, aggressive tumors such as those in the lung, breast, and pancreas are still inducing significant low survival of patients suggesting many efforts to be spent on improving the patient’s outcome [2].

Every year, about 1.5 million people in the USA and 300,000 people in the UK [3] are diagnosed with cancer. Cancer therapy and diagnosis remain the major problem in the world. In fact, a correct diagnosis is crucial for the accurate and effective treatment, because all tumors need for specific treatments such as the surgery, radiotherapy, and chemotherapy. However, to date the main problems of oncologists are the correct identification of the type of cancer, as well as the identification of the right pharmacological dose that provides the maximum therapeutic effect with the minimum toxicity [4]. In addition, there is no single treatment for the several types of cancer and each treatment has advantages and disadvantages, which often affect the patient’s quality of life [5]. Most chemotherapy drugs affect not only cancer cells but also healthy cells causing many of the symptoms related to the chemotherapy patients, including vomiting, hair loss, pale skin, etc. [6]. In this context, diagnostic approaches and chemotherapeutic delivery based on nanotechnologies, such as nanoparticles (NPs), could be promising candidates for the new era of cancer research.

In recent decades, nanotechnology dominates most of the frontier research in many fields including biomedicine [7], molecular diagnostic [8], pharmaceutic [9], optoelectronic [10], and environmental care [11]. Among the nanosystems, nanoparticles (NPs) are the most exploited. They are structures ranging from 1 to 100 nm in size that give rise to peculiar physical-chemical properties distinguished for many purposes. In the biomedical field, NPs are widely exploited for bioimaging [12], drug delivery systems [13], therapeutic agents for photodynamic (PDT) [14], photothermal therapy (PTT) [15], regenerative medicine [16,17], smart biomaterials [18], and sensing [19]. Metal nanoparticles (MNPs) are the most studied, especially noble metal NPs, including Gold (AuNPs) [20], Silver (AgNPs) [21], Platinum (PtNPs) [22] and Palladium (PdNPs) [23]. Also, polymeric [24] and lipids [25] NPs, liposomes [26], and nanohybrids represent other members of this large family.

Recently great attention has been paid to carbon-based nanomaterials such as Carbon Dots (CDs), fullerenes, Carbon nanotubes (CNTs) and Graphene Quantum Dots (GQD) that have attracted significant interest over recent years due to their excellent electrical conductivity and luminescence with broadband optical absorption and photostability, high chemical stability, low toxicity and great biocompatibility [27].

In particular, CDs are featured by a graphitic core mainly constituted of sp^2^ or sp^3^ carbon and size ranging from less than 20 nm to up 60 nm [28]. Synthetic methods for the preparation of CDs are based on two main approaches: top-down and bottom-up. In the former, CDs are obtained by breaking large carbon materials through laser ablation, acid oxidation, ultrasonic/electrochemical or hydrothermal/solvothermal exfoliation, arc discharge methods. In the bottom-up approaches, the nanostructures are prepared by carbonization of molecular precursors via microwave, hydrothermal, and thermal pyrolysis methods [29].

Quantum confinement and surface state governs the CDs properties [30] and they can be tuned by the synthetic strategy using different precursors or methods [31,32]. Actually, CDs can be designed to exhibit various functional groups including amine, carboxyl, carbonyl, hydroxyl, ether, epoxy, and heteroatoms acting also as chemical groups to graft additional materials including organic, polymeric, and biological systems. The variegated design capability to obtain several size and surface functional groups confers to these systems the possibility to modulate their chemical-physical properties, particularly the photoluminescence (PL) that exhibits a wide range of emission wavelengths as function on both size (quantum effect), surface states and groups. All these variegated properties make CDs very appealing materials in cancer application with multiple use from bioimaging to nano-carriers for drug delivery systems and promising agents for photodynamic (PDT) and photothermal therapy (PTT). In this review, we report the last evidence on the application and prospects of CDs as useful nanotheranostics tools for cancer diagnosis and therapy.

## 2. CDs Material Preparation and Properties

### 2.1. Preparation and Chemical-Physical Properties

CDs were accidentally discovered during the purification of single-walled carbon nanotubes and they are new carbon nanoparticles (about 1 to 10 nm) generally including graphitic cores surrounded by many surface functional groups [33]. CDs preparation methods are easy and inexpensive and this offers a significant advantage in nanotechnology, in terms of fast and cheap manufacture [34]. The manufacturing methods are currently divided into two main approaches that are (a) bottom-up and (b) top-down. The bottom-up method uses several heating technologies such as pyrolysis, microwave irradiation, thermal treatments to produce CDs starting from organic molecular precursor via assembly and carbonization processes (Figure 1A). On the contrary, the top-down approach uses physical technologies such as laser ablation, electrochemistry, and arc discharge to prepare smaller CDs from large carbon precursors, such as graphite, carbon fiber, carbon nanotubes, and graphene [35,36]. Recently, interesting green-chemistry approaches were proposed for the production of CDs from natural precursors [37,38], such as food and beverages [39,40,41], plants [42], solid waste derived from animal agricultural and industrial processes [43] and solid olive waste [44].

CDs’ chemical and physical properties depend on the carbonaceous core, primary composition, and surface chemical groups [45,46]. Surface groups (-COOH, -OH, -NH_2_) confer the possibility of subsequent easy functionalization with a great variety of chemical species that can then modulate the chemical-physical properties, particularly the photoluminescence. The PL arises from a variety of quantum effects that are deeply dependent on synthetic processes, such as surface imperfections, surface groups, surface passivation, optical selection of nanoparticles with different size, fluorophores with diverse degrees of π-conjugation, and heterogenous electronic states [47].

Actually, the ways to enhance CDs’ PL intensity are (a) surface modification and (b) heteroatom doping. In the first method, which conjugate CDs with molecules or polymers through covalent or non-covalent interactions, some surface energy traps become emissive due to a quantum confinement effect [32]. For the second approach, nitrogen is the most studied and efficient CDs dopant; it enhances their emission inducing an upward shift in the Fermi level and electrons in the conduction band [48].

In a typical CDs emission spectrum, the main excitation features are localized at 274 nm (π-π*) and 330 nm (n-π*) (Figure 1B). The intriguing property is the photoluminescence dependent by the excitation wavelength over the size and surface functionalization, as displayed in Figure 1C.

Actually, they typically absorb light in the UV region, but there are recent examples of visible absorption and emission in NIR (Near Infrared) range making CDs promising agents for PDT and PTT [49,50].

Therefore, due to their significant nanostructures and favorable properties, over the past decade, CDs have found application in many research areas involving light such as optoelectronic [51] and medicine [52,53]. In particular, their interesting fluorescent properties have encouraged their wide application in several fields of life science such as sensing [54,55], bioimaging [56,57], biocatalysis [58,59], and therapy [60,61,62].

### 2.2. CDs Biocompatibility

In recent years, much evidence showed that CDs are highly biocompatible in vitro. Sharma et al. [63] established the cytocompatibility of CDs prepared from Urea (CD-Urea) with dermal fibroblast showing that the treatment with CD-Urea, at the concentration of 80–200 μg/mL, considerably increase cell proliferation after 48 h. Furthermore, they also demonstrated hemocompatibility of CD-Urea (100–1000 μg/mL) in red blood cells isolated from goat blood and pro-angiogenic effect in human umbilical vein endothelial cells (HUVECs). Lu et al. [64] synthesized Nitrogen-doped carbon dots (N-CDs) and evaluated their cytotoxicity in 293T cells at several concentrations for 24 h, finding that N-CDs possess low cytotoxicity (viability > 80%) even at a concentration up to 5 mg/mL. Yan et al. [65] developed a series of CQDs carrying different surface charges. They evaluated their effects on human umbilical cord-derived mesenchymal stem cells (hUCMSCs) cytotoxicity, cellular uptake, and stability, demonstrating that relatively weak positive surface charges provide CQDs good biocompatibility. The above reported results indicate that CDs exhibit good biocompatibility over broad range of analyzed cell lines.

## 3. CDs for Bioimaging

The excellent features of CDs, such as good biocompatibility and penetrability, low toxicity, weak interactions with proteins, resistance to photobleaching, easy clearance, low cost, and easy preparation, make them a useful tool for fluorescence labeling and imaging for diagnostic applications. This section will report some studies on CDs bioimaging capabilities both in vitro and in vivo.

In the last decade, CDs demonstrated a great potential for in vitro and in vivo imaging due to their strong emission fluorescence and low cytotoxicity. In fact, due to their physical-chemical properties and size, CDs can easily penetrate biological membranes and accumulate in cell cytosol or nucleus, thus functioning as a fluorescent probe (Figure 2).

Most dyes used for fluorescence microscopy require cell fixation, and only a few are available for living cell imaging. Hua et al. [66] prepared fluorescent CDs, using a simple one-step hydrothermal treatment with the carbon sources of m-phenylenediamine and L-cysteine that can realize high-quality nucleolus imaging for not only fixed cells but also in living cells, demonstrating that these CDs possess superior properties compared with the only commercially available dye SYTO RNASelect. Ding et al. [67] incubated HeLa cells (human cervical cancer cell lines) with CDs at high concentrations (up to 5 mg/mL) for 24 h, observing both a low cytotoxic effect and an uptake only into the cytoplasm with a yellow fluorescence. Jiang group’s [68] synthesized three red, green, and blue (RGB) PL CDs and evaluated both their cytocompatibility and cell imaging. Their results showed over 90% cell viability in MCF-7 cells (human breast cancer cell line) incubated for 24 h with each of the three CDs, at concentrations from 10–50 µg/mL, and that living cells, acquired with a confocal microscope (excitation at 405 nm), fluoresced mainly in the cytoplasm, suggesting that the CDs are able to penetrate the cell membrane and enter cells. In another study, Li et al. [69] demonstrated that C-dots are biocompatible and not cytotoxic with HeLa, SMMC-7721 (human hepatocellular carcinoma cells) and HEK 293 8-Human Embryonic Kidney cells) cell lines, in concentrations up to 500 μg/mL, and that they enter into cells and are mainly confined in lysosome/endosome. Zhang et al. [70] showed that CDs prepared from polydopamine (PDA-FONs), with a concentration up to 160 mg mL^−1^, were highly biocompatible with mouse embryonic fibroblast NIH-3T3 cells (cell viability > 90%), and bright green and yellow fluorescence were mainly located in the cytoplasm. Chen et al. [71] displayed that CDs derived from carbonizing sucrose with oil acid (average 1.84 nm in size) easily penetrated the 16HBE (human bronchial epithelial cell line) but did not enter the nuclei, emitting green fluorescence around the cell membrane and cytoplasm.

Other researchers reported that CDs obtained from natural products (orange juice) did not show any cytotoxicity and were efficiently taken up by the cells (L929 and MG-63) upon incubation, exhibiting blue and green fluorescence in the cytoplasm, but not in the nuclei [72].

In addition to in vitro studies, several researchers recently evaluated the possibility of using fluorescent CDs in vivo for biomedical applications. For example, Jiang et al. [73] reported that Fluorine and Nitrogen-doped CDs (N-CDs-F), UV-Vis-NIR full-range responsive, incubated with HepG2 (human liver cancer cell line) and HeLa cells exhibit high cytocompatibility (cell viability >80%) and rapid cellular uptake in both cytoplasm and nucleus, localizing in the nucleolar region. Furthermore, they showed that N-CDs-F injected intraperitoneally into mice, at the concentration of 1 mg/mL, are able to penetrate deep tissue and so can be used for in vivo imaging. Yang et al. [74] demonstrated, for the first time that CDs maintain a strong fluorescence in vivo after subcutaneous, intradermal, and intravenous injection, as well as being highly biocompatible and non-toxic. Huang et al. [75] evaluated the effects of three fluorescent CDs injection ways (intravenous, intramuscular, and subcutaneous) on blood circulation, biodistribution, urine clearance, and passive tumor uptake, by using both near infrared fluorescence (NIRF) and positron emission tomography (PET) imaging techniques. They suggested that the injection method influences the rate of blood and urine clearance, the biodistribution of CDs in main organs and tissues, and cancer uptake over time. They also demonstrate that CDs are efficiently and rapidly uptake by the tumor when administered subcutaneously. In another study, Tao’s group [76] characterized the biodistribution of radiolabeled, photoluminescent CDs and demonstrated for the first time that they could be used for in vivo NIRF imaging. Their results suggested that CDs were slowly eliminated via the renal and fecal ways without any obvious toxic effects on animals. Licciardello et al. [77] investigated the CDs’ biodistribution and uptake in vivo through radioelement labeling and positron emission tomography (PET). Their study showed that the rapid renal clearance, biodistribution, and pharmacokinetic properties of CDs are strongly influenced by their surface charge, positive Zeta potentials, and hydrophilicity. More in detail, they demonstrated that particles with positive Zeta potentials accumulate in the liver and intestine while neutral/zwitterionic particles are rapidly cleared via the renal pathway with no significant liver uptake. The study also highlights how particle toxicity is associated with the amount of surface amine groups, as well as the possible presence of surfactant contamination traces after the CDs’ synthesis and purification. In summary, these in vitro and in vivo findings open promising scenarios for the fluorescent CDs application as tools for imaging in cancer diagnosis.

## 4. CDs for Cancer Diagnosis

The high sensitivity coupled by both the temporal and spatial resolution of fluorescence imaging make CDs one of the most promising candidates in cell target sensing and cell imaging [78]. Recently, several research groups showed that CDs with specific cell targets are able to recognize tumor cells selectively. Song et al. [79] produced CDs conjugated with folic acid (FA) (C-dots-FA) to discriminate folate receptor (FR)-positive cancer cells from normal cells (FR-negative) by culturing and analyzing a mixture of NIH-3T3 and HeLa cells. They observed that after the incubation of the cell mixture with C-dots-FA for 6 h, only HeLa cells emitted bright fluorescence, whereas NIH-3T3 cells did not, indicating that the C-dots-FA are selective only for FR-positive cancerous cells. Lee et al. [80] developed CDs conjugated with maleimide-terminated TTA1 aptamer (TTA1–CDs), highly expressed in HeLa and C6 (rat glioma cell line) but not in normal healthy CHO cells (Chinese hamster ovary cell line), and found that the incubation of the TTA1–CDs showed a strong fluorescence, selectively along cancer cell membranes and only a little uptake in normal cells. Zhang et al. [81] developed green luminescent CDs conjugated with FA (FA-CDs) and showed that the FA-CDs were able to selectively identify cancer cells within a mixture of HepG2 and PC12, displaying a bright green fluorescence, after 2 h of incubation, only in HepG2 cells. In another study, Li et al. [82] reported a new targeting tumor therapy technology based on autophagy regulation by combining the biocompatible N-doped carbon dots (N-CDs) and folic acid (FA) (FN-CDs). Their results showed that FN-CDs possess a wide range of high-targeted-ability (26 types of tumor cell lines) and affect the cellular metabolism leading to autophagy. Bhunia et al. [83] functionalized fluorescent CDs with TAT peptide or folate and incubated them with FR-positive (cancerous) and FR-negative (normal) cells, finding that TAT functionalization enhanced cell labeling and uptake and that folate selectively tagged tumor cells, which have folate receptors.

## 5. CDs for Cancer Therapy

In the last ten years, many applications of CDs in cancer theranostics were reported. Specifically, numerous researchers evaluated the possible use of CDs as targeted anticancer drug delivery [84], photodynamic therapy [85], photothermal therapy [86], as well as gene delivery [87] for cancer theranostic. This section will discuss some in vitro and in vivo applications of CDs for cancer therapy.

### 5.1. CDs for Drug and Gene Delivery

Due to luminescence, versatile surface chemistry, easy cellular internalization, and high biocompatibility, CDs are particularly interesting in drug delivery. Besides, nanoformulations also offer the possibility to increase drug solubility, bioavailability, and half-life. Several chemotherapy drugs are poorly soluble in water, not biocompatible, and have many side effects, so different CD nanocarriers were developed to avoid these drawbacks. Zheng and co-workers [88] synthesized a system based on fluorescent CDs combined with oxaliplatin derivative Oxa(IV)-COOH (Oxa-CDs) as a new treatment for metastatic colorectal cancer. They suggested that Oxa-CDs, thanks to CDs, enter cells through endocytosis, Oxa(IV)-COOH is reduced to oxaliplatin(II), reducing its toxicity to normal cells. Furthermore, although doxorubicin (DOX) is widely used for cancer treatment, it has many disadvantages, including low permeation and retention (EPR) effect, low cell internalization, and cytotoxicity to normal cells [89]. One approach to bypass these problems is to use multifunctional nanocarriers for tumor-targeted drug delivery, which have the advantage of accumulating at the tumor site due to the increased EPR effect. Yang et al. [90] constructed a nucleus-targeted drug delivery system based on the covalent conjugation of DOX and CDs functionalized with nuclear localization signal peptide (NLS-CDs) to improve its antitumor activity. They showed that the DOX-CDs delivery system can selectively accumulate in the tumor site to induce the apoptosis of A549 cells (human lung carcinoma cells) and reduce the free DOX-induced necrosis. Furthermore, they displayed that the DOX-CDs system, in vivo therapeutic studies, exhibits a higher ability to inhibit tumor growth than the free DOX. In another study, some researchers fabricated a nanocarrier for DOX-targeted delivery, using CDs capable of targeting folate receptor-positive cells conjugate with β-cyclodextrin (β-CD/CDs). Their results proved the nanocarrier’s targeting capability to folate receptor-positive cells, the intracellular uptake of DOX-β-CD/CDs, and prolonged release of DOX [91]. Wang and co-workers [92] designed a DOX delivery carrier and imaging probe for liver cancer-targeted therapy based on high fluorescence quantum yield CDs, covalently bound to folic acid (FA-CDs) and loaded with DOX (FA-CDs-D,kiOX). The in vitro investigation showed that the FA-CDs-DOX system exhibits excellent fluorescence imaging capability and can selectively deliver DOX into liver cancer cells. Furthermore, the study highlights the FA-CDs-DOX system’s fluorescence intensity was strong enough to penetrate tumor tissue and skin. The system’s targeting capacity is significantly greater than that of free DOX, as evidenced by the increased tumor inhibition. Karthik et al. [93] developed a photoresponsive drug delivery system based on a photo trigger conjugated anticancer drug (7-(3-bromopropoxy)-2-quinolylmethyl chlorambucil (Qucbl)) covalently anchored onto the surface of C-dots (Qucbl-CDs). The in vitro analysis demonstrated that the Qucbl-CDs were easily internalized into the HeLa cells and that specifically controlled the drug delivery to kill the cells upon irradiation. Wang and colleagues [94] developed a multifunctional hybrid nanocarrier composed of a Fe_3_O_4_ magnetic core, fluorescent CDs, and paclitaxel (PTX) loaded on a mesoporous silica shell. Both the in vitro and in vivo data displayed that CDs absorb and convert NIR light into heat to break PTX-mSiO_2_ bonds and release PTX, demonstrating both NIR-responsive drug release capability and the therapeutic usefulness of the combined photothermal/chemotherapy.

It has also been shown that CDs, in addition to being nanocarriers for drug delivery, can also carry genes within cells. In a recent study, Liu et al. [95] prepared a hybrid nanocarrier based on polyethyleneimine (PEI)-functionalized C-dots (CD-PEI), water-soluble and fluorescent. The in vitro assays show that CD-PEI could facilitate gene transfection in COS-7 and HepG2 cells with lower cytotoxicity as well as higher or similar efficiency compared to control PEI25k. Additionally, they display that CD-PEIs internalized into cells exhibit fluorescent emission, proposing their potential application for gene delivery and bioimaging. Wu et al. [96] developed a multi-functionalized theranostic nano-agent based on folate-conjugated reducible polyethyleneimine passivated CDs (fc-rPEICDs) complexed with siRNA molecules (fc-rPEI-CDs/siRNA). The in vitro studies displayed that the fc-rPEI-CDs/siRNA system is highly biocompatible and, after uptake, releases siRNAs in cytosol, resulting in better gene silencing and anticancer effect. More recently, Kim and co-workers [97] developed a highly biocompatible PEI-passivated CD (CD-PEI) functional siRNA delivery system and demonstrated that it can induce efficient gene knockdown both in vitro and in vivo. They found that the CD-PEI system improves siRNA cellular uptake, with low cytotoxicity, localizes in the cytoplasm and successfully releases siRNA within 12 h. They also demonstrate that the CD-PEI-siRNA delivery system can significantly silence gene expression and inhibit tumor growth in an in vivo breast cancer mouse model.

### 5.2. CDs for PDT

PDT is one of the most promising non-invasive cancer treatment approaches with limited side effects. It can be used alone or in combination with surgery, chemotherapy or ionizing radiation and can be used to destroy undetected cancerous cells at the margins of resection [98]. PDT uses photosensitizing drugs pharmacologically inactive until a particular light wavelength irradiates them in the presence of oxygen, which generates reactive oxygen species and induces cell death and tissue necrosis [99,100,101]. Figure 3 reports a schematic diagram of PDT approach.

Some photosensitizing drugs, including porphyrin and phthalocyanine derivatives, have both cancer imaging and therapy capabilities and have been approved for clinical applications [102,103]. However, their use is often limited because of reduced water solubility, photostability, prolonged cutaneous photosensitivity, and low selectivity [104]. For this region, different approaches have been investigated to combine photosensitizing drugs with other carriers such as liposomes [105], polymer nanoparticles [106,107], gold nanoparticles [108], carbon nanotubes [109], graphene’s [110,111] and carbon nanoparticles [112,113]. Recently, Huang et al. [114] prepared a novel theranostic system based on chlorin e6-conjugated C-dots (C-dots-Ce6). The in vitro results determined that C-dots-Ce6 upon irradiation exhibit good stability and solubility, low cytotoxicity, good biocompatibility, enhanced photosensitizer fluorescence detection (PFD), and remarkable photodynamic efficacy compared to Ce6 alone. Furthermore, the in vivo results suggested that the new synthesized system possess excellent imaging and tumor-homing ability without compromising the photodynamic efficacy and is effective for simultaneous PFD and PDT of cancer in vivo. In 2015, another group evaluated the in vitro and in vivo effects of a transdermal carbon dot-chlorine e6-hyaluronate (Cdot-Ce6-HA) conjugate for the photodynamic therapy of melanoma skin cancer. They showed that Cdot-Ce6-HA conjugates have a much more significant photodynamic effect on cancer cells than Ce6 and Cdot-Ce6 and that laser irradiation after topical treatment completely suppresses melanoma skin cancer [115]. Li et al. [116] prepared porphyrin-containing CDs (TPP-CDs) and proved the effective photodynamic activity in hepatoma treatment. Their results showed that TPP-CDs possess good photostability, biocompatibility, cellular uptake, and potent cytotoxicity upon irradiation in vitro and that in vivo can suppress the tumor mass. More recently, Naik et al. [117] reported synthesis, photophysical characterization, and in vitro light-induced anticancer properties of new platinum-porphyrin conjugates. Their results showed that the platinum-porphyrin conjugates display high cytotoxicity upon laser irradiation, suggesting that this porphyrin complex is a promising anticancer agent.

### 5.3. CDs for PTT

Over the last decade, PTT has also been recognized as one of the most promising cancer alternatives to more invasive conventional treatments, such as radiotherapy and chemotherapy, due to its spatial specificity and minimal invasiveness. PTT relies on photothermal agents to generate heat from laser irradiation to NIR to damage cancer cells [118]. Several nanomaterials can act as effective photothermal transducers due to their photothermal effect in the NIR region, better tumor accumulation (EPR effect), and photostability. The most used PTT nanoparticles are metallic NPs, including Gold nanorods, nanoshells, nanostars, nanocages, nano triangles, and nanoflowers [15,119]. Recently, CDs have also been used as photosensitizing agents since they can provoke substantial temperature variations under irradiation. Sun et al. [120] demonstrated that red emissive CDs (R-CDs) were able to efficiently and quickly convert laser energy into heat and that upon laser irradiation for 10 min, the viability of MCF-7 cells was significantly reduced as the R-CDs concentration increased (20–200 μg/mL). Geng et al. [50] showed that NIR-absorbing nitrogen and oxygen co-doped CDs (N-O-CDs) generate high efficiency heat under laser irradiation at low power density achieving 100% of tumor ablation without causing any side effects. Zheng et al. [121] synthesized NIR fluorescent (600 nm to 900 nm) CyCD from a hydrophobic cyanine dye and poly(ethylene glycol) (PEG800) with preferential uptake and accumulation to tumors and high photothermal conversion efficiency (38.7%) as a novel theranostic agent for NIR fluorescent imaging and PTT in vivo and in vitro. Ge et al. [122] developed CDs with intrinsic theranostic properties by using polythiophene benzoic acid (PBA) as a carbon source. They showed that the new synthesized CDs display dual photodynamic and photothermal effects under 635 nm laser irradiation, with a singlet oxygen-generating efficiency of 27% and high photothermal conversion efficiency of 36.2%. Lan et al. [123] prepared S, Se-codoped CDs by using polythiophene and diphenyl diselenide as C source and demonstrated that the doping yield excitation wavelength-independent NIR emission (peaks at 731 and 820 nm) and high photothermal conversion efficiency (58.2%) suggesting their use as new multifunctional phototheranostic agents for cancer PTT. Wang et al. [124] developed near infrared-emissive nitrogen and boron-doped CDs (B-N-CDs), with a peak at 1000 nm, to induce higher tissue penetration. Their results showed that B-N-CDs efficiently absorb and convert NIR light into heat upon laser irradiation, demonstrating a photothermal therapeutic effect that kills cancer cells in vitro and completely suppresses tumor growth in vivo.

## 6. Conclusions

Since their discovery, CDs have shown their usefulness in many research fields, especially in the biomedical applications, representing a valid alternative to traditional heavy metal-based quantum dots. In this review, we reported a summary of recent applications of CDs as innovative tools for diagnosis and therapy. Firstly, we examined their physical-chemical properties (stability and optical properties); then we analyzed their biological properties (cytotoxicity, biocompatibility, cellular internalization, and distribution); and finally, we investigated their biomedical application for cancer diagnosis and therapy (bioimaging, drug/gene delivery, PDT, and PTT). We reported some in vitro and in vivo studies on CDs, which show outstanding results regarding their cytotoxicity, biocompatibility, photostability, and anticancer effects, demonstrating that CDs could be a promising tool for biomedical applications in cancer diagnosis and therapy and potential clinical uses in the future.

## Figures and Tables

**Figure 1 cancers-13-01991-f001:**
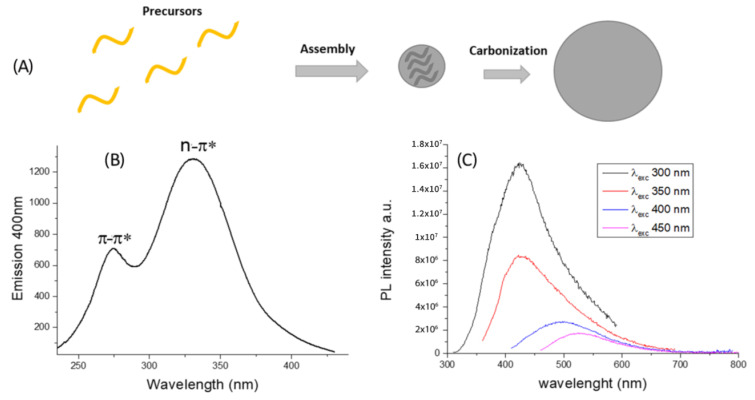
(**A**) Schematic representation of bottom-up CDs formation from molecules precursors and doping agent. (**B**) Excitation spectrum of CDs in water and (**C**) Representation of CDs wavelength-dependent photoluminescence (PL).

**Figure 2 cancers-13-01991-f002:**
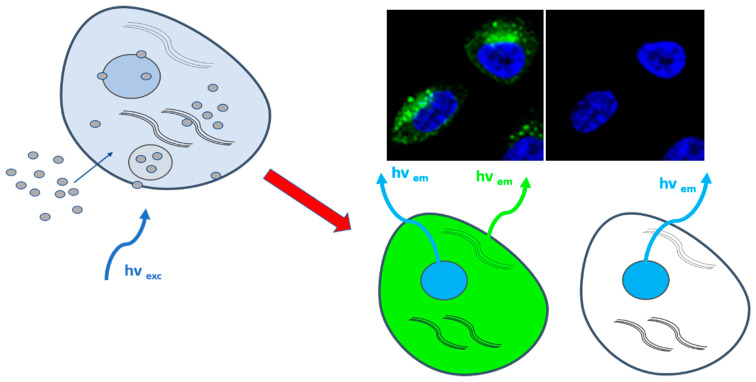
Illustration of CDs cellular internalization for bioimaging. Left part: sketch of a cell depicting CDs penetrating the cell membrane and accumulating in the cytosol (gray circles on the left); right part: sketch of representative fluorescent green staining by intracellular CDs (cell nuclei in the image are stained with DAPI (blue)).

**Figure 3 cancers-13-01991-f003:**
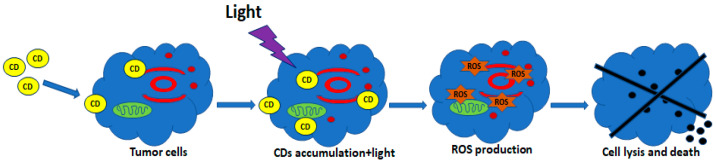
Schematic diagram of photodynamic therapy (PDT). CDs penetrate the cell membrane and accumulate in the cytosol. Light irradiation activates CDs and induces the production of reactive oxygen species (ROS) leading to the cell lysis and death.

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
