# Peer review of "Carbon Dots as Promising Tools for Cancer Diagnosis and Therapy"

_cancers, 2021, doi:10.3390/cancers13091991_

Round 1

Reviewer 1 Report

The manuscript provides a review of carbon dots, carbon-based nanomaterilas, for the applications in cancer diagnosis and therapy. The manuscript also briefly discusses the material syntheses and properties. The manuscript did thorough literature study for carbon dots. Here are some specific comments.

  1. The entire manuscript needs to be carefully revised to provide more references. For example, the manuscript discusses challenges associated with oncologist at the end of page 8/the beginning of page 9. However, the discussion does not provide any references. Another example is the discussion about carbon-dot properties at page 9/line 79.
  2. The manuscript has many one-sentence paragraphs. It’s not clear whether these are simple editing issues. For example, page 9/line 88, line 91, and line 95.
  3. If Figure 1 and 2 are from the published papers, the manuscript needs to provide references.
  4. Figure captions need more description. For example, the schematic diagram and images in Figure 2 are not self-explanatory.
  5. (minor issue) Page 10/line 106 has two “[”
  6. (minor issue) Figure 3 is a schematic diagram.

Author Response

The manuscript provides a review of carbon dots, carbon-based nanomaterials, for the applications in cancer diagnosis and therapy. The manuscript also briefly discusses the material syntheses and properties. The manuscript did thorough literature study for carbon dots. Here are some specific comments.

  1. The entire manuscript needs to be carefully revised to provide more references. For example, the manuscript discusses challenges associated with oncologist at the end of page 8/the beginning of page 9. However, the discussion does not provide any references. Another example is the discussion about carbon-dot properties at page 9/line 79.

Answer:

We thank the reviewer for the comment. As suggested, we added some references (4,5; 30-32; 55-60) both in the text and in the references section.

  1. The manuscript has many one-sentence paragraphs. It’s not clear whether these are simple editing issues. For example, page 9/line 88, line 91, and line 95.

Answer:

Thank you for the comment. As suggested by the reviewer, we modified one-sentence paragraphs, since these were simple editing issues.

  1. If Figure 1 and 2 are from the published papers, the manuscript needs to provide references.

Answer:

Thank you for the comment. Both Figure 1 and 2 are new made by the authors and they aren’t from published papers.

  1. Figure captions need more description. For example, the schematic diagram and images in Figure 2 are not self-explanatory.

Answer:

Thank you for the comment. As suggested, we modified and integrated both figure 2 and 3 captions.

  1. (minor issue) Page 10/line 106 has two “[”

Answer:

Thank you for the comment. We eliminated “[“.

  1. (minor issue) Figure 3 is a schematic diagram.

Answer:

Thank you for the comment. As suggested we modified “schematic representation” with “schematic diagram” in the text.

Reviewer 2 Report

CDs possess strong emission fluorescence, low cytotoxicity. weak interactions with proteins, resistance to photobleaching, easy clearance, low cost, and easy preparation. They have found applications in cancer diagnosis and therapy in recent years. The review considers various combinations of multimodal nanoparticles with enhanced receptor properties to increase the selectivity of interaction with target cells. The review on CDs biomedical use is extremely relevant and will be of interest to many readers. The manuscript is well written and can be accepted for publication.

Remarks

fields including biomedicine [5], molecular diagnostic [6] pharmaceutics [7], optoelectron-55

Have to be

fields including biomedicine [5], molecular diagnostic [6], pharmaceutics [7], optoelectron-55

Author Response

CDs possess strong emission fluorescence, low cytotoxicity. weak interactions with proteins, resistance to photobleaching, easy clearance, low cost, and easy preparation. They have found applications in cancer diagnosis and therapy in recent years. The review considers various combinations of multimodal nanoparticles with enhanced receptor properties to increase the selectivity of interaction with target cells. The review on CDs biomedical use is extremely relevant and will be of interest to many readers. The manuscript is well written and can be accepted for publication.

Remarks

fields including biomedicine [5], molecular diagnostic [6] pharmaceutics [7], optoelectron-55

Have to be

fields including biomedicine [5], molecular diagnostic [6], pharmaceutics [7], optoelectron-55

Answer:

Thank you for the comment. We modified the text according to the remarks.

Reviewer 3 Report

This work comprehensively describes carbon dots and their applications in cancer therapy including drug and gene delivery, PDT, PTT, as well as biomedical imaging. Several suggestions can enhance the potential of this paper making it more attractive to the reader:

  • I suggest to proofread the paper with a native speaker as, although the Gemma is mostly intact, stylistically the text is subpar and also requires some editing of biological terminology.
  • Since authors aim to show both therapeutic and diagnostic applications, the application of CDs to cancer detection should be discussed, potentially, even as a separate section.

Author Response

This work comprehensively describes carbon dots and their applications in cancer therapy including drug and gene delivery, PDT, PTT, as well as biomedical imaging. Several suggestions can enhance the potential of this paper making it more attractive to the reader:

  • I suggest to proofread the paper with a native speaker as, although the Gemma is mostly intact, stylistically the text is subpar and also requires some editing of biological terminology.

Answer:

Thank you for the comment. As suggested we proofread the paper with a biologist expert.

  • Since authors aim to show both therapeutic and diagnostic applications, the application of CDs to cancer detection should be discussed, potentially, even as a separate section.

Answer:

Thank you for the comment. As suggested we separated the two sections as follow:

  1. CDs for bioimaging
  2. CDs for cancer diagnosis
